# Visualising Combined Time Use Patterns of Children’s Activities and Their Association with Weight Status and Neighbourhood Context

**DOI:** 10.3390/ijerph16050897

**Published:** 2019-03-12

**Authors:** Jinfeng Zhao, Lisa Mackay, Kevin Chang, Suzanne Mavoa, Tom Stewart, Erika Ikeda, Niamh Donnellan, Melody Smith

**Affiliations:** 1School of Nursing, The University of Auckland, Auckland 1023, New Zealand; n.donnellan@auckland.ac.nz (N.D.); melody.smith@auckland.ac.nz (M.S.); 2School of Sport and Recreation, Auckland University of Technology, Auckland 0627, New Zealand; lisa.mackay@aut.ac.nz (L.M.); tom.stewart@aut.ac.nz (T.S.); erika.ikeda@aut.ac.nz (E.I.); 3Department of Statistics, The University of Auckland, Auckland 1010, New Zealand; k.chang@auckland.ac.nz; 4Melbourne School of Population and Global Health, The University of Melbourne, Melbourne 3010, Australia; suzanne.mavoa@unimelb.edu.au

**Keywords:** time use, accelerometer data, physical activity, sedentary behaviour, sleep, neighbourhood context, weight status, school children, compositional analysis, visualisation

## Abstract

Compositional data techniques are an emerging method in physical activity research. These techniques account for the complexities of, and interrelationships between, behaviours that occur throughout a day (e.g., physical activity, sitting, and sleep). The field of health geography research is also developing rapidly. Novel spatial techniques and data visualisation approaches are increasingly being recognised for their utility in understanding health from a socio-ecological perspective. Linking compositional data approaches with geospatial datasets can yield insights into the role of environments in promoting or hindering the health implications of the daily time-use composition of behaviours. The 7-day behaviour data used in this study were derived from accelerometer data for 882 Auckland school children and linked to weight status and neighbourhood deprivation. We developed novel geospatial visualisation techniques to explore activity composition over a day and generated new insights into links between environments and child health behaviours and outcomes. Visualisation strategies that integrate compositional activities, time of day, weight status, and neighbourhood deprivation information were devised. They include a ringmap overview, small-multiple ringmaps, and individual and aggregated time–activity diagrams. Simultaneous visualisation of geospatial and compositional behaviour data can be useful for triangulating data from diverse disciplines, making sense of complex issues, and for effective knowledge translation.

## 1. Introduction

Time is a finite and scarce resource that shapes people’s daily lives as they schedule various activities. How people use their daily time is a major determinant of people’s well-being and health [1,2]. Seen from a perspective of movement, one’s daily living is made up of a sequence of activities such as moderate-to-vigorous physical activity (MVPA; generally >3 metabolic equivalents (METs), e.g., brisk walking or running), light-intensity physical activity (LPA; 1.5–3 METs, e.g., casual walking), sedentary behaviour (SB; ≤1.5 METs), and sleep. Activity can be accrued through a range of dimensions, including organised sport, unstructured free play, and active travel (i.e., walking, cycling, or scootering to destinations). In particular, active travel to school has received much attention in the last decade, given the low proportion of children who actively travel to school despite established benefits to child health and the environment.

Numerous studies have shown that increased intensity of physical activity, adequate sleep time, and reduced SB are associated with better health outcomes [3,4,5,6,7,8,9,10,11,12]. For example, Chastin, et al. [9] found that re-allocating 10 min of SB to MVPA led to a waist circumference reduction of 0.001%. Similarly, when time on SB or LPA were reallocated for MVPA, lower adiposity and higher cardiorespiratory fitness were predicted, although the changes were small [13]. WHO [3] has promoted four strategic objectives to prioritise physical activity as a regular part of people’s everyday life. These include creating active societies, active environments, active people, and active systems.

Traditionally, this body of research has examined activities in isolation from each other, or with little adjustment for time spent on other behaviours [6,7]. Given a 24-h time budget each day, time spent on one activity has an impact on the availability of time for other activities, meaning that daily activities are collinear and co-dependent. Indeed, Chaput, et al. suggested that conducting one “unhealthy” activity can moderate the health benefits of another [7], and that the benefits of MVPA might be reduced if children have poor sleep habits and/or engage in excessive SB [8]. Similarly, preventing a transfer of time from LPA to SB might reduce the negative effects of physical inactivity on obesity [9]. Using a time-use survey, Dunton et al. [10] found a joint association of physical activity and SB with body mass index (BMI) in US adults. For example, there was a weaker association between leisure-time MVPA and BMI among those spending large amounts of time watching TV and movies, and vice versa. Given their compositional nature, it is important to investigate daily activities in their entirety [6,7,8,9,11,12]. The new Canadian 24-h Movement Guidelines for Children and Youth [14] are a tangible reflection of the movement towards understanding the composition of daily activity for health, despite a still-emerging evidence base.

Childhood obesity has become a subject of worldwide concern [15]. In New Zealand, the prevalence of obesity in children has reached epidemic proportions. One in nine NZ children aged 2–14 years (11%) were obese in 2014/2015, and rates were much higher for Māori (15%) and Pacific children (30%) and those living in the most deprived areas [16]. Childhood obesity negatively impacts the current health of children, and also influences their future health. For example, it is associated with increased risk of diabetes, arterial hypertension, coronary artery disease, and fatty liver disease in later life [17]. Therefore, there is a need for innovative research into potential strategies to reduce the significant personal, economic and structural impacts of obesity [18,19].

Childhood obesity is a complex disorder affected by many risk factors, including lower socioeconomic status and area-level socioeconomic deprivation. Evidence suggests that the prevalence of childhood obesity is strongly correlated with socioeconomic status and is highest among children living in the most deprived areas [18]. A study by Singh et al [20] found that, between 2003 and 2007, obesity prevalence increased by 23% to 33% for US children in low-education, low-income, and high-unemployment households and overweight prevalence increased by 13% to 15% in low-education and low-income households. In the same time period, no significant increases in obesity or overweight prevalence were observed for children in other socioeconomic groups. Furthermore, there was a growing class gap in obesity rates between US adolescents aged 12–17 from upper and lower socioeconomic status backgrounds and the prevalence of obesity among young people with upper socioeconomic status has decreased in recent years, whereas it has continued to increase among their low-socioeconomic-status peers [21]. At the neighbourhood level, Singh et al [22] found that US children living in the least favourable social conditions such as unsafe surroundings, poor housing, and no access to sidewalks, parks, and recreation centres had 20–60 percent higher odds of being obese or overweight than children not facing such conditions, and studies in the UK [23,24] found that the prevalence of overweight and obese children was positively associated with area deprivation. The existence of a gender gradient was demonstrated by Kinra, et al. [24] who found that the odds ratio (OR) for childhood obesity was quite large and significant for girls (OR: 1.39, 95% confidence intervals (CI): 1.08 to 1.80, *p* = 0.011) but was not significant for boys (OR: 1.29, 95% CI: 1.00 to 1.65, *p* = 0.049).

Most time-use studies in the health field investigate the duration of daily activities, but the timing of activities is also important for health outcomes such as weight status. Patel and Spaeth [25] found that adults (aged 22–70 years) in the US with obesity, relative to normal-weight individuals, were significantly less likely to be sleeping on weekdays between 21:30 and 5:30, and on weekends/holidays from 22:30 to midnight, but were significantly more likely to be sleeping on weekdays between 08:00 and 10:00, and 13:00 and 15:00, and on weekends/holidays between 12:00 and 16:00. Such approaches can help identify critical windows for supporting health-promoting behaviours; however, to date, the evidence base remains limited. Accelerometer data are increasingly being used to assess patterns of human daily activities [26,27,28]. These data also provide valuable timing information, which was only available from time-use diaries in the past. The growing availability of such data provides both opportunities and challenges for turning information into knowledge. There is a trade-off between investigating complex time-use behaviour patterns and presenting results that are easy to understand.

Visualisation—the graphical display of abstract information—has been widely recognised as an essential, intuitive, and thought-provoking tool for sense-making (also called data analysis) and communication [29]. The human eye and brain can extract and interpret information on a map more rapidly than the most elegant software [30]. The power of visualisation relies on its ability: (1) to augment human visual ability in perceiving, detecting, and exploiting highly complex structures and patterns [31]; (2) to arouse the imagination for exploration; and (3) to provide instruments and toolsets for the exploration of knowledge and insights [32].

Visualisation has been extensively employed in health geography research, for example to explore spatial patterns of BMI [33], multivariate epidemiological data [34,35], and multimorbidity [36]. A socio-ecological approach recognises that an individuals’ behaviours can be significantly impacted by external factors, such as the environment in which he/she lives [37]. In the context of activity and health, infrastructural (e.g., activity-friendly landscapes) and sociodemographic factors (e.g., area-level deprivation) are key factors of relevance [24,38]. Integrating geographical data with 24-h activity composition is a novel concept that holds great promise for gaining a comprehensive understanding of the behaviour, timing, and broader context in which health-related behaviours occur.

### Aims and Rationale

The aim of this research is to visualise compositional time-use patterns of children’s daily activities in relation to their weight status and area-level deprivation using accelerometer data collected from the Neighbourhoods for Active Kids project [26].

An innovative open source visualisation tool called a time–activity diagram was developed for this research using R software [39]. Time–activity diagrams present the timing and duration of activities at both individual and aggregated levels. They provide immediate insights into the significance of children’s time-use and activity trajectories and their association with weight status and neighbourhood context. These visualisations, combined with inventive use of ringmap techniques developed by the first author [35,40], integrate space, time, activity, neighbourhood deprivation, and weight status together, provide insights that would otherwise be inaccessible.

## 2. Methods

### 2.1. Participants and Data

#### 2.1.1. Participants

Participants in the study were children aged 8–13 years from 10 primary (these in school years 5 and 6) and 9 intermediate schools (those in school years 7 and 8) in Auckland, the largest city in New Zealand. Figure 1 shows the flow of participants through the selection, eligibility checking and data cleansing processes. At the end of that process, information for 882 children was included in this study.

#### 2.1.2. Data

Participants were given Actigraph GT3X+ accelerometers fixed to an elastic belt (Actigraph, Pensacola, FL, USA) worn around the waist for 7 days. Using Actilife v6 software, the devices were initialised to log raw data at 30 Hz. After download, all data were collapsed into 30-s epochs in preparation for analysis. Accelerometer-derived, time-stamped activity intensity information (i.e., SB, LPA, and MVPA) was categorised using Evenson’s cut-points [41]. Sleep and waking were recorded by the child’s parent/caregiver in a daily diary. Children’s biological sex (binary variable of male/female) was reported by the child’s parent/caregiver during a computer-assisted telephone interview survey [26]. Objectively assessed height and weight were used to derive BMI and weight status (underweight, normal, overweight, and obese) using Cole 2012 International Obesity Task Force (IOTF) cut-offs [42].

The home locations of participants were collected using an internet-based survey tool called softGIS, and these locations were used to measure the shortest distance between home and school along the pedestrian transportation network [43]. The home locations of 10 children that were not recorded in the softGIS survey were sourced from geocoded home addresses provided by the children’s parents. Children’s regular travel mode (classified as active travel to school or passive travel to school) was also captured in the softGIS survey.

The children’s home locations were aggregated to Data Zones (a customised geography for health and social research with populations ranging from 500 to 1000) [44,45] to protect privacy and to display the geographical distribution of participants in association with deprivation. The New Zealand Indices of Multiple Deprivation (IMD, [44]) were linked to children’s home locations to provide neighbourhood-level socioeconomic information. The IMD includes 28 indicators, which represent seven domains of deprivation: employment. Income, crime, housing, health, education, and geographical access.

The analysis was limited to children who had complete information on their age, height, weight and home address; and weekdays were days on which accelerometer data were available for seven or more hours [46] spent on three activity types: MVPA, LPA, and SB; and at least five hours sleep time were recorded.

The accelerometer data were analysed using SAS Enterprise Version 7.1 (SAS Institute Inc.: Cary, NC, USA) [47], and the proportion of each activity was then calculated for weekdays using total time spent on MVPA, LPA, SB and sleep as the denominator.

### 2.2. Visualisation Strategies

#### 2.2.1. Using Ringmaps to Show an Overview of Patterns in the Data

Ringmaps offer an innovative, data-driven, and compact way of visualising space, time and activity in association with their contextual settings [48]. The ringmap is made up of a geographical map in the centre and surrounding rings. The circular configuration of the rings is especially suitable for representing cyclic time such as 24 h days [40]. Rings can be divided into sectors to represent variables such as sequential time blocks, and each ring can represent an abstract spatial area, for example, most deprived Data Zones or areas within 800 m of schools.

In this research, the ringmap is made up of five rings, each representing a quintile of IMD deprivation ordered from the least deprived areas, quintile 1, in the innermost ring to the most deprived areas, quintile 5, in the outermost ring. Each ring is divided into eight sectors, representing eight 3-h time blocks ordered clockwise from midnight at the top. In each time block, the average percentage of time spent on each activity is displayed using proportionately sized colour dots. The inset map shows the geographical distribution of IMD by Data Zone as a background map overlaid with the aggregated locations of children’s home addresses.

#### 2.2.2. Using Small-Multiple Ringmaps to Compare Patterns in Sub-Sets of the Data

Small multiples are matrices of small sized graphic representations that support understanding and comparison of complex information [49]. They normally use the same form of representation and are useful for comparing related patterns [50]. Small multiples can be ordered in a meaningful way to help readers perceive trends and relationships in the data and to gain insights [51]. Well-designed small multiples will reveal the most thematically relevant key features, and allow viewers to inspect the visualisation at their own pace and in their preferred viewing order, avoiding issues commonly associated with non-interactive animations, e.g., viewers have a limited ability to grasp detailed information and keep it active in their working memory [52].

In this research, four small-multiple ringmaps based on the ringmap in Section 2.1.1 are showcased to compare activity patterns in association with deprivation by sex and weight status. The small multiples are shown in a 2 (columns) by 2 (rows) matrix. In the matrix, the first row shows patterns for boys, and the last row shows patterns by girls. The two columns show weight status with those classified as underweight or normal weight on the left and classified as overweight or obese on the right.

#### 2.2.3. Developing Time–Activity Diagrams to Visualise Patterns at Both the Individual and Aggregated Levels

An innovative open source visualisation tool, called a time–activity diagram, was developed using R software [39]. Two types of diagram were created.

The first diagram presents the timing of activities for two groups of children: (1) children who were classified as underweight or normal weight and (2) children who were classified as overweight or obese at the aggregated level. The diagram consists of two panels: the aggregated patterns for the two groups are shown together in the panel on the left, while the mean percentage differences are shown on the right. To create the aggregated information, each activity was summarised at 10-min intervals. The aggregated mean percentage of time spent on each activity for each group is presented using bandwidth (the four activities are summed to 100% for each individual) and activities were overlaid where they occurred simultaneously with semi-transparent colours showing the different activity types. The mean percentage difference in time spent on each activity between the two weight status groups was calculated as the mean percentage for the underweight/normal weight group minus the mean percentage for those classified as overweight or obese. Therefore, any results that appear on the right hand side of the “0” axis indicate that the percent of time spent on that activity was greater among underweight/normal weight children, and vice versa.

The second diagram presents the timing and sequencing of activities at the individual and the aggregated levels. It consists of two panels, with children who were classified as underweight or normal weight in the left panel and children who were classified as overweight or obese on the right panel. At the individual level, the diagram embeds information about each child’s activity and timing in one vertical line, and the lines are ordered by a variable of interest, for example distance from home to school. Aggregated information, created using a method similar to the previous visualisation, for the two groups of children appears at opposite ends of the two panels.

## 3. Results

### 3.1. Socio-Demographic and Weight Status Characteristics of Participants

Table 1 shows characteristics of participants by age, gender, weight status, and area-level deprivation. There were 437 boys and 445 girls. Girls were slightly over-represented in the youngest group (55.6%), more were classified as overweight or obese (53.7%), and more lived in the most deprived areas (54.0%) compared to boys; 75.4% of the children were classified as normal or underweight, while 24.6% were classified as overweight or obese.

### 3.2. Ringmap Overview

Figure 2 shows overall spatial and temporal activity patterns over 24 h on weekdays for all 882 children, and their association with area-level deprivation. The inset map shows that participants’ neighbourhood deprivation was heterogeneous across the sample.

As expected, time spent sleeping (purple dots) fully occupies the first two time blocks (00:00 to 06:00) and dominates the last time block (21:00 to midnight). The percentage of time spent in LPA (light green) and SB (orange dots) dominates the four time blocks from 09:00 to 21:00, with the time spent in LPA greatest during school hours. The percentage of time spent in MVPA (dark green colour) is the smallest, barely exceeding 7%.

There are deprivation-related differences in the percentage of time spent in different activities. For example, between 21:00 and midnight, children residing in the most deprived areas sleep the least (they have the largest negative difference in percentage activity compared to those living in the least deprived areas so they have the largest purple ring) and spend more time in SB and LPA (with the large blue rings). Between 09:00 and 12:00, children in IMD quintiles 4 and 5 spent relatively more time in SB (large blue rings) and less time in LPA (large purple rings).

### 3.3. Small-Multiple Ringmaps

Figure 3 shows a small-multiple ringmap matrix that enables comparisons of spatial, temporal, and activity patterns over 24 h on weekdays by sex, weight status, and area-level deprivation. The top and bottom rows of the matrix represent boys and girls, respectively. The left column represents boys and girls classified as underweight or normal weight, while the right column represents boys and girls classified as overweight or obese.

The inset geographical maps show that the home locations of children with overweight or obese status were more dispersed than those of children with underweight or normal status. The ringmaps in the left column (Figure 3a,c) show that the strongest clustering of children who were classified as underweight or normal weight occurred in Central and North Auckland where there is less deprivation, while the ringmaps in the right column (Figure 3b,d) show that the strongest clustering for children who were overweight or obese occurred in deprived areas in South Auckland.

In the time block from 21:00 to 00:00, boys and girls classified as underweight or normal weight in the most deprived areas slept less (purple rings) and spent significantly more time in LPA and SB (blue rings) in comparison to their counterparts in the least deprived areas. The same pattern was found for overweight or obese boys residing in the most deprived areas, but not for girls. In the time block from 09:00 to 12:00, boys in both weight status groups in the most deprived areas spent more time in LPA and less time in SB in comparison with boys residing in the least deprived areas, but there was no deprivation gradient for girls.

### 3.4. Time–Activity Diagrams for Aggregated Patterns

In Figure 4, a subset of the data, namely activities on weekdays for boys in the most deprived areas, was used to showcase a time–activity diagram. In the left panel, time spent in MVPA peaked during the school break times. SB dominates the time period between 17:00 and 20:00, especially for boys classified as overweight or obese (the orange bandwidth is greater than the other colours).

In the right panel, we see that boys classified as being underweight or normal weight spent more time doing LPA during school hours and early evening (i.e., their results appear on the right-hand side of the zero axis). They also slept earlier in the evening. Boys classified as overweight or obese spent more time in SB throughout the day, especially in the period immediately after school finished at 15:00.

### 3.5. Time–Activity Diagrams for Individual and Aggregated Patterns

Figure 5 displays activity patterns at both the individual and aggregated levels for the two weight status groups for boys in the least deprived areas on Wednesdays and/or Thursdays. We limited the scope to two days to reduce the number of individual timelines in the visualisation. The individual activity patterns (shown as timelines in the middle of the visualisation) give us a sense of what the original data look like and reflect the structure of the weekday for both groups. For example, MVPA and LPA occur during breaks in the weekday and just before and after school. Sleep started from as early as 19:00 for some boys.

Aggregated patterns for the two weight status groups are shown on the far left and right, and are the same as those in the left panel in Figure 4. Here, the combination of both individual and aggregated information provides a fuller picture.

## 4. Discussion

Traditional approaches that report aggregated data commonly miss the timing of activities and the composition of an individual’s daily activity patterns, which are complex and may differ in meaningful ways in different populations. It is now becoming accepted that daily activities should be analysed and conceptualised within a compositional paradigm to achieve meaningful findings [6,9,12]. Studies that have revealed significant associations between time use, physical activity, and children’s health include Olds, et al. [1], who found that screen time and school-related time were the most elastic activities in Australian adolescents. Every additional hour committed to physical activity was associated with 32 min less screen time, and screen time reduction was more pronounced in obese adolescents (−56 min) compared with in normal (−31 min) and overweight (−27 min) adolescents. This research provided evidence for healthy behavior intervention. Dumuid, Stanford [53] reinforced the importance of MVPA to children’s health using a compositional isotemporal model. They suggested that while interventions to increase MVPA may be of benefit, it is important to avoid a decline in MVPA levels, particularly among already inactive children.

However, the results of most compositional analysis are presented using statistical approaches alone, which are not easily understood by a general (i.e., non-academic or non-specialist) audience. Visualisation of time-use data offers a useful means of understanding nuanced patterns of behaviour, for example by identifying key intervention times for different groups. Examples include Olds, et al. [54] who used a Chord diagram to show aggregated in-and-out time flows between activity categories such as Work, Chores, and Transport in Australia, Vrotsou, et al. [55] who visualised daily activity paths of individuals to search for sequencing patterns in Sweden, and Zhao, et al. [40] who used ringmaps and a time geography approach to reveal spatial, temporal, and activity patterns in legacy time-diary data in Halifax, Canada. In the United Kingdom, histomaps were used to show how time use in a standard weekday has changed between 1961 and 2015 [56], and a transitional flow diagram was used to show how children’s physical activity changed between ages 6 and 9 [57]. Cumulatively, these visualisation techniques provided insights into time-use and activity patterns in an intuitive way. However, it is rare to see visualisations of time use that expose the complex composition of activity patterns, especially at a fine temporal scale.

The unique contributions of this research are the development of time–activity diagrams and the novel utilisation of ringmap techniques for visualising compositional activity patterns over time, space, weight status, and neighbourhood deprivation at both the individual and aggregated levels.

Ringmaps are compact, data-driven, and an ideal form for representing cyclic time, for example, 24 h days. They are suitable for representing spatial, temporal, and activity information at the population or sub-population level, as shown in this paper. They can also be applied at the individual level [58] for a small sample of people. The use of time blocks coupled with compositional activities represents a new way of using ringmaps.

Time–activity diagrams are an innovative way to compositionally visualise and compare activities in linear time at a fine granularity at both the individual and aggregated levels. The centrally located semi-transparent time–activity “rods” (see the left panel in Figure 3, and both ends in Figure 4) facilitate easy comparison between groups and within a group over time and compositional activities. These diagrams provide a unique opportunity to show both the “forest” and the “trees” and this can inspire immediate insights. These visualisation techniques were showcased using accelerometer data, but they can easily be applied to other kinds of data that involve space, time, and activity/behaviour, such as time-diary data. These techniques would be ideal for visualising clinical data that change over circadian time, such as blood pressure. Time––activity diagrams was developed using Open Source R software and the R code was published at figshare [59] to facilitate replication, adaptation, and development of this technique.

The innate strengths of visualisation can be harnessed by using appropriate visualisation strategies. In this research, we used a “top-down” strategy to visualise our data. The ringmap overview provided a high-level visualisation showing the entire data in a compositional and aggregated form in the context of cyclic time. We then used small multiples with the same ringmap configuration to split the data into four subsets (boys, girls, and two weight statuses) for comparison. Having seen the overview and the subsets, we then presented detailed information using time–activity diagrams to explore individual and aggregated patterns in linear time at a fine scale. This research is exploratory. While its focus was on demonstrating new visualisation techniques and strategies, rather than discovering new knowledge, the patterns revealed in our showcase visualisations are consistent with previous studies suggesting face validity of the use of this technique as applied in this study. For example, the peaks of MVPA that are associated with school breaks and before and after school during weekdays are in line with the time distribution of MVPA revealed by Olds, et al. [60]. Across all children, reduced physical activity levels and increased SB during the post-school period demonstrate the potential for interventions to increase activity during the “critical window” for physical activity.

Our ringmap overview visualisation revealed less desirable time-use behaviours in children from the most deprived areas. For example, they slept later in the evening and were more sedentary during morning school hours in comparison to their peers in the least deprived areas. These findings contribute to knowledge about the relationship between deprivation, time-use patterns of activities and childhood obesity. Understanding the context of activity is an important step forward in understanding the composition of daily activities in children. For example, in Australian children aged 10–13 years, qualitative research revealed numerous factors that impacted their physical activity engagement during the critical window—including neighbourhood safety, distance between destinations, and weather [61]. These variables could easily be integrated into the technique presented in this paper to identify differences in activity composition by objectively derived environmental factors (e.g., replacing deprivation with a measure of neighbourhood destination accessibility). We also identified new, and more nuanced critical time windows for intervention. For example, in the time block from 09:00 to 12:00, boys who were classified as overweight or obese and lived in the most deprived areas had less healthy time-use behaviours (less time spent on MVPA and LPA and more time spent on SB) compared to those in the least deprived areas. While in the time block from 21:00 to 24:00, boys and girls who were classified as underweight or normal and lived in the most deprived areas slept less and spent more time on LPA and SB.

Nevertheless, this research is not without its limitations. For example, neighbourhood deprivation information was represented as abstract ring space. It may be challenging for readers to navigate between ring space and geographical space, even with the assistance of an inset geographic map. Interactively linked visualisations may mitigate this issue [62]. While fine grain information is useful in representing data at their original granularity, the individual-level time–activity diagram can only effectively display a few hundred individuals in a full extent on a normal-sized computer screen. To represent information for a large number of individuals, an aggregated time–activity diagram is recommended.

We used simple arithmetic means for time spent on each activity to showcase the visualisation techniques in this paper. However, these techniques can be used in combination with statistical analyses such as compositional data analyses. For example, one could use the difference between compositional means which have been expressed as isometric log ratios [6] to detect differences in time use between groups, and then intuitively visualise (and hopefully understand) these differences using these visualisation techniques.

Children’s sleep times were reported by their parents, so these data may involve memory and proxy-report bias. For the ringmap visualisations, each participant only provided one line of data, which was derived by calculating mean values for each child, irrespective of whether they contributed one or more days’ worth of data. This could mean that within-subject variability could be greater or less for the data presented here, so findings related to differences between groups should be interpreted with care.

Active travel to school has an important association with physical activity and health-related fitness [63,64], but it was not the focus in this paper. Mode of travel to school information was used as a variable to group data to showcase visualisations. In our ongoing research, we would examine travel mode in greater detail. In addition, mode of travel in this research was only related to weekdays, which are quite structured and lend themselves to strong, distinct patterns. One would anticipate that activities undertaken on weekends are far less structured since people’s time is free. This limitation should be kept in mind when discussing patterns observed between different weight status groups. How weekend activities are associated with children’s body size and neighbourhood context could be the subject of further research.

The accelerometer data we collected did not provide associated geographical position at each time point, so our time–activity diagrams do not have a spatial dimension. With the increasing availability of accelerometer data and the ease with which locational information can be collected with devices such as Global Positioning System [65], a time–activity–location diagram will be needed. One solution would be to integrate time geography visualisation approaches [40,48,58] with time–activity diagrams.

## 5. Conclusions

This research developed innovative visualisation techniques and strategies that intuitively represent compositional activities for children over various time scales in association with their weight status and neighbourhood level deprivation as a whole. The preliminary results from these showcase visualisations suggest that there are critical time windows for intervention both during school hours and after school.

## Figures and Tables

**Figure 1 ijerph-16-00897-f001:**
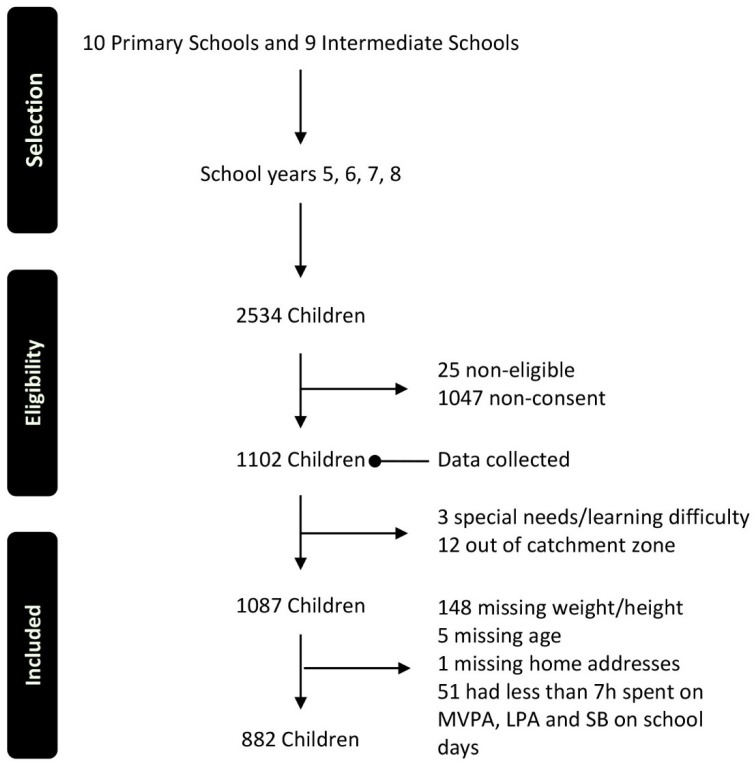
Flow of selection, eligibility checking, data cleansing, and study inclusion. Moderate-to-vigorous physical activity (MVPA); light-intensity physical activity (LPA); sedentary behaviour (SB).

**Figure 2 ijerph-16-00897-f002:**
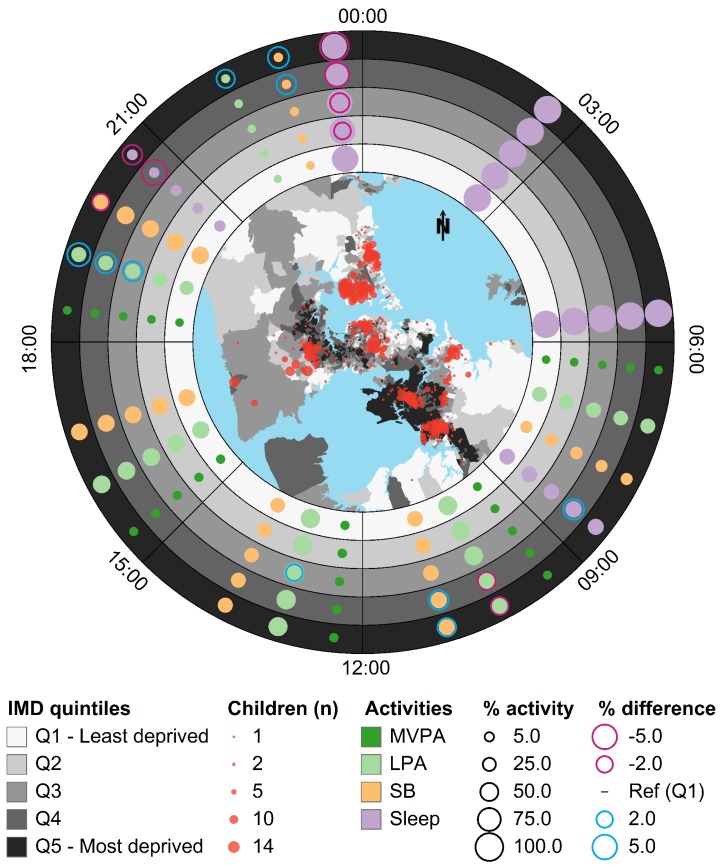
Ringmap showing overall spatial and temporal activity patterns over 24 h on weekdays for all children (*N* = 882) and their association with deprivation. The geographic units in the inset map are Data Zones and proportional semi-transparent dots in red represent aggregated number of children who lived in each Data Zone. Each ring in the ringmap represents a quintile of IMD deprivation, which is divided into eight 3-h time blocks ordered clockwise from midnight at the top. White to black shades represent IMD quintiles from 1 (the least deprived) to 5 (the most deprived) in both the inset geographic map and the 5 rings. In each time block, the average percentage of time spent on each activity is displayed using proportional coloured dots. Dark green represents MVPA, light green represents LPA, orange represents SB, and purple represents sleep. To facilitate comparison between the same activity in the same time block across the five IMD quintiles, proportional circular outlines representing negative and positive differences in average percentage of time spent on each activity compared to the least deprived quintile (% difference = 0%) are overlaid. Percentages of less than 1% and differences in percentages between −2% and 2% are suppressed.

**Figure 3 ijerph-16-00897-f003:**
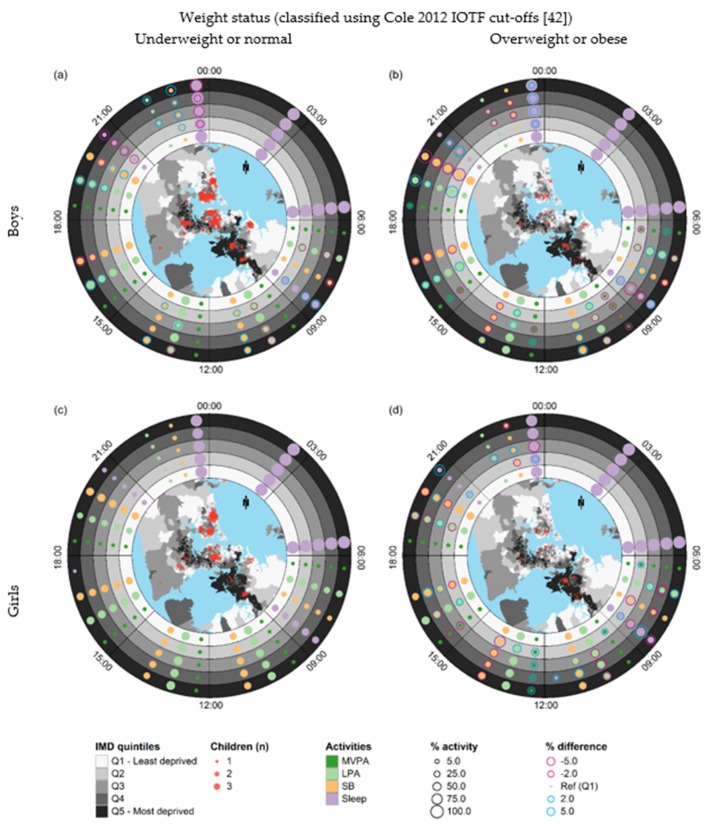
Small-multiple ringmaps comparing time-use patterns on weekdays for children who regularly actively travelled to school in association with deprivation by sex and weight status: (**a**) boys with underweight or normal status; (**b**) boys with overweight or obese; (**c**) girls with underweight or normal status; (**d**) girls with overweight or obese status. The configuration of each ringmap is the same as the ringmap in Figure 2. Proportional circular outlines representing negative and positive differences in average percentage of time spent on each activity compared to the least deprived quintile (% difference = 0%) are overlaid. Percentages of less than 1% and differences in percentages between −2% and 2% are suppressed.

**Figure 4 ijerph-16-00897-f004:**
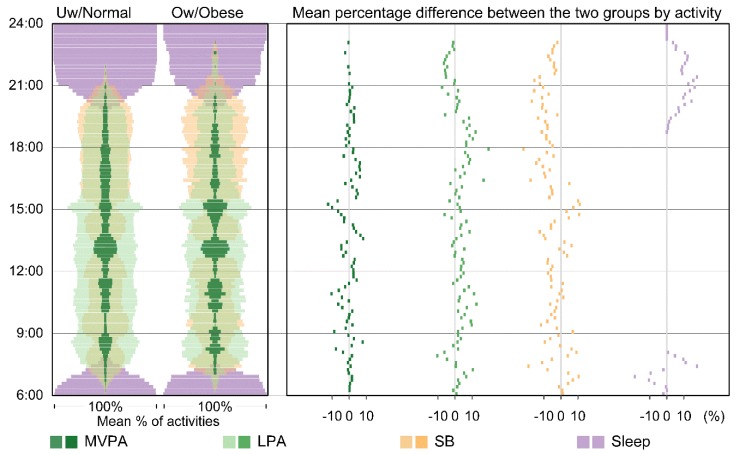
Time–Activity diagram showing aggregated time-use patterns of activities for the two weight status groups in the least deprived areas. The vertical dimension of the visualisation shows time of day from 6:00 at the bottom to midnight at the top. Activities are aggregated at 10-min intervals. The percentages of time in each interval are summed to 100 percent. The left panel shows the aggregated mean percentage of time spent on the different activities for the two groups. The mean percentage of time is presented as a proportional bandwidth. The right panel shows percentage differences between the two groups for each activity type. The percentage difference was calculated by subtracting the mean percentage of time for the overweight or obese group from that of the underweight or normal group (i.e., the underweight or normal group was the reference). Any results that appear on the right-hand side of the zero axis (positive values) indicate that the percentage of time spent on that activity was greater among underweight or normal children. Colours represent the four activity types: MVPA (dark green), LPA (light green), SB (orange), and sleep (purple). The colours in the left panel are semi-transparent in order to facilitate comparison of different compositional activity patterns both within each weight status group and between the two weight status groups.

**Figure 5 ijerph-16-00897-f005:**
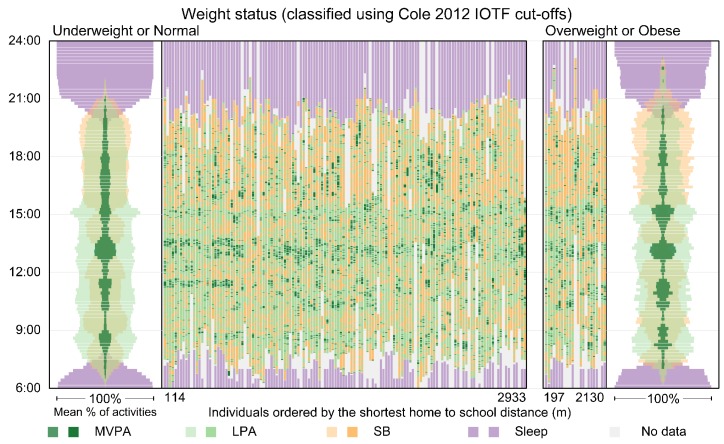
Time–Activity diagram showing individual and aggregated patterns using data for Wednesdays and/or Thursdays for boys in the least deprived areas who actively travelled to school. The diagram is configured as 2 main panels with boys who were classified as underweight or normal on the left and boys who were classified as overweight or obese on the right. The vertical dimension shows time of day. Individual-level time-use patterns for the 4 activity types for the two groups are shown in the middle of the diagram. Each vertical timeline represents one boy’s sequential daily activities from 6:00 to midnight, with the light grey colour indicating missing information. The boys’ timelines are ordered within each panel by travel distance from home to school in meters. Note that an active travel classification indicates the regular travel mode. It does not guarantee that the child actually travelled to school actively on the two days displayed here. Travel distances greater than 3000 m were removed.

**Table 1 ijerph-16-00897-t001:** Socio-demographic and weight status characteristics of study participants.

	All	Boys		Girls	
	*n*	*n*	%	*n*	%
Total	882	437	49.5	445	50.5
*Age group*		(mean = 10.66, SD = 1.18)		(mean = 10.53, SD = 1.20)	
≤9	187	83	44.4	104	55.6
10	234	115	49.1	119	50.9
11	234	119	50.9	115	49.1
≥12	190	101	53.2	89	46.8
Weight		(mean = 43.24, SD = 12.17)		(mean = 43.70, SD = 14.31)	
Height		(mean = 1.49, SD = 0.09)		(mean = 1.48, SD = 0.10)	
*Weight status (classified using Cole 2012 IOTF cut-offs* [42])					
Underweight or Normal	640	325	50.8	315	49.2
Overweight or Obese	242	112	46.3	130	53.7
*IMD Quintiles*		(mean = 2.89, SD = 1.48)		(mean = 3.0, SD = 1.5)	
Q1—Least deprived	194	101	52.1	93	47.9
Q2	204	101	49.5	103	50.5
Q3	157	79	50.3	78	49.7
Q4	112	57	50.9	55	49.1
Q5—Most deprived	215	99	46	116	54

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
