# Peer review of "Visualising Combined Time Use Patterns of Children’s Activities and Their Association with Weight Status and Neighbourhood Context"

_ijerph, 2019, doi:10.3390/ijerph16050897_

Round 1

Reviewer 1 Report

I would like to congratulate the authors on an interesting, easy to read paper that will provide a useful visualization tool for the emerging field of time-use epidemiology.

I have a few minor suggestions.

Abstract:

Line 22: the daily time-use composition must be outlived by everyone on this planet, so I don’t think it makes sense for it to be promoted or hindered. Perhaps the authors are referring to the health implications of the daily time-use composition?

Line 23: It seems a bit odd to say “this research has developed”, when it is the researchers that are doing the developing.

Line 24: consider replacing “to” with “and”… “over a day and generate new…”

Line 25-27. This needs to be split into two sentences for ease of reading

P1 Introduction:

Line 35: consider reordering “time daily” to “daily time”

P2. Line 45. References are required.

Line 62. Missing the word “in”… “obesity in children”

Line 75: what do you mean by this? They don't capture the full 24 hours, and don't capture enough days?

P3. Line 100. The study name is mentioned 3 times throughout the paper, which gets a bit repetitive. I suggest removing it from here.

Line 102. This is the first time R is mentioned, so you need to cite or insert city, date…

Line 101-104. Can this get put into two sentences for ease of reading?

Line 106: replace “which integrates” with “integrate”

Line 118-119 replace “where there was available accelerometer” with something like “where accelerometry was available”.

Line 120: five hours sleep time was recorded… replace “was” with “were”

Line 120-121: as it is currently written, you don’t specify how you determined what was valid information.

Line 125: as it is currently written, it appears that only participants that were missing all three of these were missing. Maybe replace “and” with “and/or”.

Line 136: reference for IMD?

P4. Line 159: proportionately-sized?

P13. Line 387: specify that they were “arithmetic” means.

Line 389-390: I find the wording a bit confusing here. It sounds like compositional means are isometric log ratios. Perhaps rephrase to something like: “find the compositional difference between compositional means” or “find the difference between compositional means which have been expressed as isometric log ratios”.

In summary, I really enjoyed looking at the visualizations presented and can see that they will be a useful tool for the interpretation of complex data.

Author Response

Comments and Suggestions for Authors

I would like to congratulate the authors on an interesting, easy to read paper that will provide a useful visualization tool for the emerging field of time-use epidemiology.

I have a few minor suggestions.

Dear reviewer, thank you very much for the constructive suggestions.  They have helped us to improve the quality of our paper greatly. Please see our point by point answers below.

 Abstract:

Line 22: the daily time-use composition must be outlived by everyone on this planet, so I don’t think it makes sense for it to be promoted or hindered. Perhaps the authors are referring to the health implications of the daily time-use composition?

Response 1: we have updated the wording to ‘the health implications of the daily time-use composition’ as the reviewer suggested. 

Line 23: It seems a bit odd to say “this research has developed”, when it is the researchers that are doing the developing.

Response 2: we have modified the words to ‘we developed…’.

Line 24: consider replacing “to” with “and”… “over a day and generate new…”

Response 3: we have replaced “to” with “and” as the reviewer suggested. Thank you.     

Line 25-27. This needs to be split into two sentences for ease of reading

Response 4: we have split the sentence into two sentences as the reviewer suggested.

P1 Introduction:

Line 35: consider reordering “time daily” to “daily time”

Response 5: we have reordered “time daily” to “daily time” as the reviewer suggested.                       

P2. Line 45. References are required.

Response 6: we have added references as the reviewer suggested.

Line 62. Missing the word “in”… “obesity in children”

Response 7: we have added the word “in”. 

Line 75: what do you mean by this? They don't capture the full 24 hours, and don't capture enough days?

Response 8: we have removed the sentence. 

P3. Line 100. The study name is mentioned 3 times throughout the paper, which gets a bit repetitive. I suggest removing it from here.

Response 9: we have removed the study name as the reviewer suggested.        

Line 102. This is the first time R is mentioned, so you need to cite or insert city, date…

Response 10: we have added the reference.  

Line 101-104. Can this get put into two sentences for ease of reading?

Response 11: we have split the sentence into two sentences as the reviewer suggested. 

Line 106: replace “which integrates” with “integrate”

Response 12: we have replaced “which integrates” with “integrate” as the reviewer suggested. 

Line 118-119 replace “where there was available accelerometer” with something like “where accelerometry was available”.

Response 13: we have replaced “where there was available accelerometer” with “where accelerometer data were available”.  

Line 120: five hours sleep time was recorded… replace “was” with “were”

Response 14: we have replaced “was” with “were” as the reviewer suggested.  

Line 120-121: as it is currently written, you don’t specify how you determined what was valid information.

Line 125: as it is currently written, it appears that only participants that were missing all three of these were missing. Maybe replace “and” with “and/or”.

Response 15: for the above two comments (Line 120-121, Line 125), we have clarified information as follows:

The analysis was limited to children who had complete information on their age, height, weight and home address; and weekdays where accelerometer data were available for seven or more hours [33] spent on three activity types: MVPA, LPA and SB; and at least five hours sleep time were recorded.”

Line 136: reference for IMD?

Response 16: we have added the reference.   

P4. Line 159: proportionately-sized?

Response 17: we have adopted the reviewer’s suggestion. 

P13. Line 387: specify that they were “arithmetic” means.

Response 18: we have replaced “mathematical means” with “arithmetic means” as the reviewer suggested. 

Line 389-390: I find the wording a bit confusing here. It sounds like compositional means are isometric log ratios. Perhaps rephrase to something like: “find the compositional difference between compositional means” or “find the difference between compositional means which have been expressed as isometric log ratios”.

Response 19: we have modified the sentence as follows:

“For example, one could use the difference between compositional means which have been expressed as isometric log ratios [6] to detect differences in time use between groups, then intuitively visualise (and hopefully understand) those differences using these visualisation techniques.”. 

In summary, I really enjoyed looking at the visualizations presented and can see that they will be a useful tool for the interpretation of complex data.

Submission Date

21 January 2019

Date of this review

12 Feb 2019 02:46:37

THANK YOU!

Best regards,

Jinfeng

Reviewer 2 Report

Dear Zhao et al

it was a distinct pleasure to review this manuscript, which presented a novel, refreshing and scientifically sound addition to the literature. Of particular strength is the results section, and I applaud I feel this will make a significant addition to the literature, however I would request a few edits, detailed below, prior to acceptance of this manuscript. 

Abstract -

on the whole the abstract was fine, but I would request the addition of basic participant information into the 'method' portion of the abstract. This helps the reader understand what is to come.

Introduction - 

line 37 - "huff and puff" and other similar descriptors - please avoid the overly colloquial descriptors, rather, use physiologically sound descriptions - in this case, what about METs or oxygen uptake or even crude accelerometry counts. I appreciate the verbatim description, but feel it is out of place.

line 44 - "...increased physical activity" - this statement/sentence needs expansion - i.e. increased PA from what? by what magnitude of increase etc

line 47 - please expand and provide some examples of such policy and practice influences 

line 65 - "....future health" - expand on how future health is impacted?

line 67 - please add citations to this effect

Methods - 

please include a section on "participants" (or other wording of your choice) - I realise you have referred the reader to the NfAK, but some basic description is required here to aid the reader, without having to locate additional studies .

likewise, within this section basic mean and SD information for (e.g.) height, weight, age, SES etc is needed. 

I also feel that the accelerometer description (set up, processing and analysis) is a little light on detail.-- granted it is not the main purpose of the study, but further detail is required in line with physical activity research.

2.2 visualisation strategies -- im not convinced this requires a small sub-section of its own. I would think it is more pragmatic to either subsume with the subsequent paragraph, or just remove

otherwise, the methods is soundly written and presented

results - 

the results are, in my opinion, excellent. a very enjoyable read. 

discussion - 

line 329 - "inspirational" - I prefer removal of such colloquial terms 

the content of the presented discussion is solid, I do request that the authors make some "additions", rather than edits - again, although the main aim is around the visualisation technique, I feel a paragraph on the implications of time-use for childrens health and PA is missing, authors like Tim Olds and Dorothea Dumuid do a good job of doing this. I don't think it requires substantive discussion, but some discussion nonetheless. I think paying some homage to the work of e.g. Olds and Dumuid, among others, how their time-use analysis compares.

I also feel a brief presentation of limitations is warranted to give a broader picture to the reader. 

**overall comments, - check some of the terminology throughout, I found some descriptions to be more colloquial than scientific

aside from these minor suggestions, I welcome the publication of this work, and look forward to being able to cite!

best wishes with your revision and continued work

Dr Cain Clark 

Author Response

Comments and Suggestions for Authors

Dear Zhao et al

it was a distinct pleasure to review this manuscript, which presented a novel, refreshing and scientifically sound addition to the literature. Of particular strength is the results section, and I applaud I feel this will make a significant addition to the literature, however I would request a few edits, detailed below, prior to acceptance of this manuscript. 

Dear Dr Cain Clark, thank you very much for the constructive suggestions.  They have helped us to improve the quality of our paper greatly. Please see our point by point answers below.

Abstract -

on the whole the abstract was fine, but I would request the addition of basic participant information into the 'method' portion of the abstract. This helps the reader understand what is to come.

Response 1: we have added basic participant information as follows as you suggested:

“The 7-day behaviour data used in this study were derived from accelerometer data for 882 Auckland school children and linked to weight status and neighbourhood deprivation.”

Introduction - 

line 37 - "huff and puff" and other similar descriptors - please avoid the overly colloquial descriptors, rather, use physiologically sound descriptions - in this case, what about METs or oxygen uptake or even crude accelerometry counts. I appreciate the verbatim description, but feel it is out of place.

Response 2: we have removed “huff and puff” wording and added relevant METs as you suggested.

line 44 - "...increased physical activity" - this statement/sentence needs expansion - i.e. increased PA from what? by what magnitude of increase etc

Response 3: We modified "...increased physical activity" to "...increased intensity of physical activity (e.g., from LPA to MVPA)…"

We also added following sentence:

“For example, Chastin, et al. [9] found that re-allocating 10 minutes of SB to MVPA led to a waist circumference reduction of 0.001%. Similarly, when time on SB or LPA were reallocated for MVPA, lower adiposity and higher cardiorespiratory fitness were predicted, although the changes were small [1].”

line 47 - please expand and provide some examples of such policy and practice influences 

Response 4: we have added following sentences:

“WHO [3] has promoted four strategic objectives to prioritise physical activity as a regular part of people’s everyday life. These include creating active societies; active environments; active people; and active systems.”

line 65 - "....future health" - expand on how future health is impacted?

Response 5: we have expanded it as follows:

“Childhood obesity negatively impacts the current health of children, and also influences their future health. For example, it is associated with increased risk of diabetes, arterial hypertension, coronary artery disease, and fatty liver disease in later life [16].”

line 67 - please add citations to this effect

Response 6: we have added citations.

Methods - 

please include a section on "participants" (or other wording of your choice) - I realise you have referred the reader to the NfAK, but some basic description is required here to aid the reader, without having to locate additional studies .

Response 7: we have added a section on participants as you suggested.

likewise, within this section basic mean and SD information for (e.g.) height, weight, age, SES etc is needed. 

Response 8: we have provided basic mean and sd information for age, gender, weight, height and deprivation in Table 1.

I also feel that the accelerometer description (set up, processing and analysis) is a little light on detail.-- granted it is not the main purpose of the study, but further detail is required in line with physical activity research.

Response 9: we added more information on accelerometer description as you suggested.

2.2 visualisation strategies -- im not convinced this requires a small sub-section of its own. I would think it is more pragmatic to either subsume with the subsequent paragraph, or just remove

Response 10: we have removed the small paragraph as you suggested.

otherwise, the methods is soundly written and presented

results - 

the results are, in my opinion, excellent. a very enjoyable read. 

discussion - 

line 329 - "inspirational" - I prefer removal of such colloquial terms 

Response 11: we have removed the word “inspirational” as you suggested.

the content of the presented discussion is solid, I do request that the authors make some "additions", rather than edits - again, although the main aim is around the visualisation technique, I feel a paragraph on the implications of time-use for childrens health and PA is missing, authors like Tim Olds and Dorothea Dumuid do a good job of doing this. I don't think it requires substantive discussion, but some discussion nonetheless. I think paying some homage to the work of e.g. Olds and Dumuid, among others, how their time-use analysis compares.

Response 12: we have added a paragraph as follows as you suggested:

“Studies that have revealed significant associations between time use, physical activity and children’s health include Olds, et al. [8], who found that screen time and school-related time were the most elastic activities in Australian adolescents. Every additional hour committed to physical activity was associated with 32 minutes less screen time, and screen time reduction was more pronounced in obese adolescents (-56 minutes) compared with normal (-31 minutes) and overweight (-27 minutes) adolescents. This research provided evidence for healthy behavior intervention. Dumuid, et al. [9] reinforced the importance of MVPA to children’s health using a compositional isotemporal model. They suggested that while interventions to increase MVPA may be of benefit, it is important to avoid a decline in MVPA levels, particularly among already inactive children.”

I also feel a brief presentation of limitations is warranted to give a broader picture to the reader. 

Response 13: we have added a limitation paragraph as follows:

“Nevertheless, this research is not without its limitations. For example, neighbourhood deprivation information was represented as abstract ring space. It may be challenging for readers to navigate between ring space and geographical space, even with the assistance of an inset geographic map. Interactively linked visualisations may mitigate this issue [10]. While fine grain information is useful in representing data at their original granularity, the individual level time-activity diagram can only effectively display a few hundred individuals in a full extent on a normal-sized computer screen. To represent information for a large number of individuals, an aggregated time-activity diagram is recommended.”

**overall comments, - check some of the terminology throughout, I found some descriptions to be more colloquial than scientific

Response 14: we have checked the paper throughout, thank you.

aside from these minor suggestions, I welcome the publication of this work, and look forward to being able to cite!

best wishes with your revision and continued work

Dr Cain Clark 

Submission Date

21 January 2019

Date of this review

08 Feb 2019 12:00:57

THANK YOU!

Best regards,

Jinfeng